# Are *Melanoides tuberculata* and *Tarebia granifera* (Gastropoda, Thiaridae), suitable first intermediate hosts of *Clonorchis sinensis* in Vietnam?

Hung Manh Nguyen [1,2]*, Hien Hoang Van[1], Loan Thi Ho[1], Yulia V. Tatonova[3], Henry Madsen[4]

1 Institute of Ecology and Biological Resources (IEBR), Vietnam Academy of Science and Technology (VAST), Hanoi, Vietnam, 2 Graduate University of Science and Technology, Vietnam Academy of Science and Technology (VAST), Hanoi, Vietnam, 3 Federal Scientific Center of the East Asia Terrestrial Biodiversity, Far Eastern Branch, Russian Academy of Sciences, Vladivostok, Russia, 4 Parasitology and Aquatic Diseases, Department of Veterinary Disease Biology, Faculty of Health and Medical Sciences, University of Copenhagen, Copenhagen, Denmark

* hung_iebr@yahoo.com, nmhung@iebr.ac.vn

**Data Availability Statement:** All data underlying the findings are fully available without restriction.

## Abstract

### Background

Two thiarid snail species, *Melanoides tuberculata* and *Tarebia granifera* have been reported as first intermediate hosts of the liver fluke *Clonorchis sinensis*; however, their role as true first intermediate hosts has not been verified. Thus, the present study aimed to clarify the suitability of these two snail species as first intermediate hosts of *C. sinensis*. This was accomplished by collecting snails from a highly endemic area for *C. sinensis* in Vietnam, the Thac Ba reservoir, and identifying shed cercariae using molecular techniques. We also conducted experimental infections of five snail species including *M. tuberculata* and *T. granifera* with eggs of *C. sinensis*.

### Methodology/Principal findings

A total of 11,985 snails, representing 10 species were sampled. Five snail species, *M. tuberculata*, *T. granifera*, *Lymnaea swinhoei*, *Parafossarulus manchouricus*, and *Bithynia fuchsiana* were found shedding cercariae with an overall prevalence of infection ranging from 0.7% to 11.5%. Seven cercarial types were recorded. Cercariae of *C. sinensis* were only found in *Parafossarulus manchouricus*. Using a multiplex PCR approach for detecting *C. sinensis* infection, the prevalence in *P. manchouricus* was 4.2%. Additionally, all five snail species were experimentally exposed to *C. sinensis* eggs, however only *P. manchouricus* was successfully infected with an infection rate of 7.87%.

### Conclusions/Significance

We confirmed that in the Thac Ba reservoir, Vietnam, the two thiarids, *M. tuberculata* and *T. granifera* are not suitable first intermediate hosts of *C. sinensis*. Only *P. manchouricus* was

All relevant data are within the paper and its Supporting Information files.

**Funding:** This research is funded by Vietnam National Foundation for Science and Technology Development (NAFOSTED) under grant number 106.05-2018.17. The funders had no role in study design, data collection and analysis, decision to publish, or preparation of the manuscript.

**Competing interests:** The authors have declared that no competing interests exist.

found infected by *C. sinensis* in nature, and was the only species that became infected experimentally.

## Authors' summary

Currently 13 snail species are reported as first intermediate hosts of *Clonorchis sinensis*, including two species of the Thiaridae, *Melanoides tuberculata* and *Tarebia granifera*. Both snail species have wide distributions in tropical and subtropical waters across the World, while the distribution of *C. sinensis* is much narrower and only occurs in endemic areas of East Asia. The role that these two thiarid snail species plays as successful hosts of *C. sinensis*, however, has been refuted in other studies. The present study was conducted to clarify this issue. Based on the results from field surveys in a newly discovered and highly endemic area for *C. sinensis*, as well as, data obtained under experimental conditions, we confirm that in Northern Vietnam, the two thiarid snail species are not successful first intermediate host of *C. sinensis*. *C. sinensis* was only detected in a single snail species, *Parafossarulus manchouricus*, through observation, molecular analyses and experimental infections.

## Introduction

One of the most important fish-borne zoonotic trematodes is *Clonorchis sinensis* (Cobbold, 1875) (Opisthorchiidae), the small liver fluke that parasitizes humans and fish-eating mammals throughout Northeast Asia [1–5]. This trematode species causes serious disease in humans and animals, including; pericholangitis, pyelophlebitis, cholangiohepatitis and multiple abscesses [3,4].

Thirteen freshwater snail species have been reported as first intermediate hosts of *C. sinensis* (Table 1). A majority of these determinations, 10 of the 13, are based solely on morphological examination of cercariae (pleurolophocercous type). Only two species, *Alocinma longicornis* and *Parafossarulus manchouricus*, are confirmed first intermediate hosts through experimental infection and molecular diagnostics [6,7] and an additional snail species, *Bithynia fuchsiana* has been confirmed by molecular data alone [8].

*Melanoides tuberculata* and *T. granifera* are highly invasive species and are currently present in North (southern USA, Mexico), Central (Panama, Caribbean region) and South America (Venezuela, Brazil), Europe, Africa, Australia, Pacific islands and Asia [1,24,25]. Mas-Coma & Bargues [1] mentioned that their wide distribution potentially could result in the global spread of *C. sinensis*. However, the role of thiarid snails as first intermediate hosts of liver flukes has been questioned by Bui et al. [26] and Nguyen et al. [27], as they did not find cercariae of *C. sinensis* within thousands of specimens of thiarid snails collected from areas known to be highly endemic for clonorchiasis in Vietnam. In addition, Besprozvannykh et al. [28] experimentally exposed fish to all cercarial types shed from *M. tuberculata* collected from Northern Vietnam and found no *C. sinensis* metacercaria. Similarly, in China, Zhang et al. [12] found *C. sinensis* larvae within four snail species, i.e. *P. manchouricus*, *P. sinensis*, *B. fuchsiana* and *A. longicornis*, while *M. tuberculata* was not infected.

The present study aimed to verify the role of *M. tuberculata* and *T. granifera* as first intermediate hosts of *C. sinensis* through multiple approaches, i.e. field surveys, experimental infection, and molecular detection of *C. sinensis*.

**Table 1. List of the first intermediate hosts for *C. sinensis*.**

| Snail species | Locality [Reference] |
|---|---|
| **Assimineidae** | |
| *Assiminea lutea* Adams, 1861 | China [9,10] |
| **Bithyniidae** | |
| *Alocinma longicornis*[*],[**] (Benson, 1842) (synonym: *Bithynia longicornis*) | China [8,11−14] |
| *Bithynia chaperi* Morlet, 1886 | Vietnam [1,12] |
| *B. fuchsiana*[**] Moellendorff, 1894 | China [8,11−14] |
| *B. misella* Gredler, 1884[~] | China [11,13,14], Vietnam [15] |
| *B. siamensis* Lea, 1856 | Vietnam [16] |
| *Parafossarulus anomalospiralis* Liu, Li & Liu, 1985 | China [11,14,17] |
| *P. manchouricus*[*],[**] (Bourguignat, 1860) (synonym: *P. striatulus*) | China [8,11,12,14], Japan [1,18], Korea [1,19], Russia Federation [1,2,7], Vietnam [20,21] |
| *P. sinensis* Faust, 1930 | China [12,14] |
| **Semisulcospiridae** | |
| *Semisulcospira cancellata* (Benson, 1833) | China [14,22] |
| *S. libertina* (Gould, 1859) | China [11] |
| **Thiaridae** | |
| *Melanoides tuberculata* (Müller, 1774) | China [23], Vietnam [20,21] |
| *Thiara granifera* (Lamarck, 1822) | Taiwan [11] |

Note

[*] Confirmed by experimental infection

[**] Confirmed by molecular data; [~] failed in an experimental infection.

## Materials and methods

### Ethics statement

All applicable international, national, and institutional guidelines for the care and use of animals were followed. Euthanasia of laboratory animals was carried out in accordance with the Committee on the Ethics of Animal Experiments of Federal Scientific Center of the East Asia Terrestrial Biodiversity, Russia (Permit Number: 3 of 02.06.2011).

### Study area, snail sampling and examination

Thac Ba reservoir covers an area of 23,400 ha of Yen Bai Province in the mountainous region of Northern Vietnam. There are 12 ethnic groups living in communes around the reservoir, and local people usually prepare and eat raw fish from the reservoir [29]. According to recent studies, the Thac Ba reservoir is known as a highly endemic area for *C. sinensis* [29,30]. In this region, the sewage system of each household is very simple and domestic waste water and latrines flush directly into canals nearby which flow into the reservoir. Domestic animals (e.g. cattle, pigs, ducks, chicken, dogs, cats) are free roaming during the day and locked up during the evening and nights. Manure from domestic animals is likely to enter the reservoir. The coexistence of intermediate hosts (snails and fishes) and the habit of eating raw or insufficiently cooked fish explain the high prevalence of fish-borne zoonotic trematodes in this region.

Ten sites within the reservoir were selected for snail collection, including 2 sites at Gia Binh, 2 sites at Thac Ba, and 6 sites at Vu Linh commune (Fig 1). Snails were collected every four months from April 2015 to April 2019. During morning hours, snails were collected by

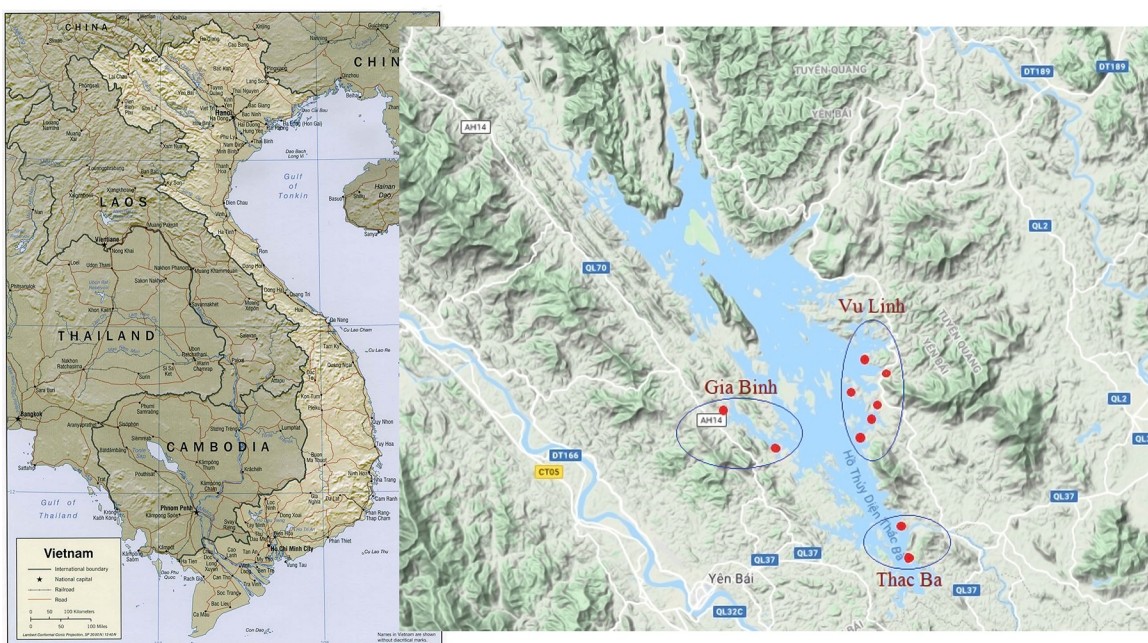

**Fig 1. Study sites (red dots) in the Thac Ba reservoir.**

hand for 30 minutes per site. Snails were transferred to plastic containers and transported alive to the laboratory where they were identified following the keys of Brandt [31] and Dang [32]. All snails were recorded and checked for trematode infection using the shedding method. Depending on the size of each snail, each individual was placed into a small plastic container with 5 or 10 ml of water each and left for 48 hours to shed cercariae. Snails were checked each day for shedding at 8:00 AM, 1:30 PM and 5:30 PM. Cercariae of *C. sinensis* were identified based on their morphological characters [7] and they were taken randomly from each infected snail to confirm their identity using molecular technique (see below). Cercariae of other trematode species were identified to major types according to the keys of Ginetsinskaja [33], Shell [34] and other available references.

## Experimental infection of snails

Five snail species were chosen for experimental infection, including 4 species from Vietnam: *M. tuberculata*, *T. granifera*, *B. fuchsiana*, *P. manchouricus* and 2 species from Russia *Koreoleptoxis amurensis* and *P. manchouricus*. For experimental infection, snails were collected from trematode-free locations of the suburb of Hanoi, Vietnam and Primorsky Region, Russia. Snails were acclimatized to laboratory conditions for seven days and fed dried lettuce leaves.

For maintenance of experimental snails, aquaria with a bottom layer of sand and small stones were filled (plastic boxes with 2L capacity) with decholorinated tap water. To mimic the natural habitat of these snails, a few pieces of brushwood (*Terminalia catappa*) and aquatic plants (*Hygrophila difformis*, *Lemna minor*) were added to each aquarium. Ten individuals of the same snail species were housed in each aquarium. The number of each snail species used in this experiment varied.

For collecting the *C. sinensis* samples from Vietnam, livers and gall bladders of 7 cats were purchased in a slaughter house in Nam Dinh province where restaurants serving cat meat are popular (even if it is technically illegal) [35]. After that, they were placed on ice and carried

fresh to the laboratory, the duration for transporting samples were 1.5 hours. In the laboratory, samples were necropsied immediately to find trematodes. The *C. sinensis* samples from Russia were collected from experimentally infected rats. A total of 87 adult specimens of *C. sinensis* from Vietnam and 15 specimens from Russia, were kept alive in saline at 37 $^{\circ}$C for egg release. After 6 hours, the uteri of all *C. sinensis* specimens were opened and eggs collected. Eggs were placed in 20 g of feces collected from parasite free domestic cats and 700 ml of tap water was added and shaken until the fecal matter disintegrated. A multichannel pipette (8 channels) with 1 mL tips was used to transfer 5 ml of fecal-egg suspension to each aquarium for experimental infection. When drawing up the suspension, the tip of the pipette reached to the bottom of the glass flask containing the suspension to make sure that eggs are present in the aquaria. After 3 weeks, living snails in each aquarium were recorded and fixed in absolute ethanol for molecular analyses.

## Multiplex PCR for detecting *C. sinensis* larvae in snails

Snail specimens were individually crushed. Shell fragments and the muscular foot of the snail were removed, and only the hepatopancreas, which was easily freed from the remaining soft tissue, was kept for extracting genomic DNA. The DNA samples were extracted using the Hot-SHOT technique [36]. The ITS1 rDNA region of *C. sinensis* and the 28S rDNA gene of the snails were co-amplified using two sets of primers designed using the Oligo Calc. program (http://biotools.nubic.northwestern.edu/OligoCalc.html), including: CSF (5'–TGTTCTACA TGTATGTTCCGC–3', forward), CSR (5'–TAGCTCGAGACACATCATGC–3', reverse), for ITS region; 28SF (5'–AGTAACGGCGAGTGAAGCG–3', forward), 28SR (5–CTACCGAGCT GAATTCCGC–3', reverse) for 28S region. The PCR reaction mixture (20μl of total volume) contained 1μl (5 pM) of each primer, 10 μl of Hot Start Green-Taq Master Mix (2 ×) (Promega Corporation, USA) and approximately 10 ng of total DNA. PCR was performed on an Eppendorf Mastercycler using the following cycling conditions: a 5 min initial denaturation step at 95˚C, 38 cycles of 25 sec at 95˚C, 25 sec at 57˚C and 35 sec at 72˚C, and a 5 min extension at 72˚C. The PCR products were separated by electrophoresis using a 2% agarose gel stained with RedSafe (cat # 21141, iNtRON).

Negative controls (PCR mix with $H_2O$ instead of DNA) were included, along with positive controls (20 ng DNA of *C. sinensis* and 20 ng DNA of non-infected snails). The optimal concentration of DNA for both the snail and *C. sinensis* was evaluated using two different concentrations: 10 ng of snail DNA and 100 ng of *C. sinensis* DNA; 100 ng of snail DNA and 10 ng of *C. sinensis* DNA.

PCR products were purified by ExosapIT enzymatic cleanup and sequenced using Sanger sequencing method. The four primers CSF, CSR, 28SF and 28 SR mentioned above were used for sequencing.

## Results

### Natural trematode infections

A total of 11,985 snails, representing 10 species, were collected (Table 2). Among them, 11,237 snails were examined for cercarial shedding. Five snail species, including *M. tuberculata*, *T. granifera* (Thiaridae), *P. manchouricus*, *B. fuchsiana* (Bithyniidae) and *L. swinhoei* (Lymnaeidae), were found shedding cercariae. The prevalence of infection in snails ranged from 0.7% to 11.5%.

A total of seven cercariae types were recorded. Cercariae of *C. sinensis* belong to the pleurolophocercous group and can be identified from others within the group by distinct morphological characteristics and their overall movement. In Table 3, we arranged separately *C*.

**Table 2. Prevalence of infected snails from Thac Ba reservoir.**

| Snail species | Examined for actively shedding | | Examined for *C. sinensis* by PCR | |
|---|---|---|---|---|
| | No. snails examined | No. infected snails (%) | No. snails examined | No. snails infected (%) |
| **Thiaridae** | | | | |
| *Melanoides tuberculata* | 2,013 | 151 (7.5) | 265 | 0 |
| *Tarebia granifera* | 288 | 33 (11.5) | 116 | 0 |
| *Thiara scabra* | 85 | 0 | – | – |
| **Viviparidae** | | | | |
| *Angulyagra polyzonata* | 3,386 | 0 | – | – |
| *Sinotaia aeruginosa* | 675 | 0 | – | – |
| **Lymnaeidae** | | | | |
| *Lymnaea swinhoei* | 426 | 3 (0.7) | – | – |
| **Ampullariidae** | | | | |
| *Pomacea canaliculata* | 107 | 0 | – | – |
| **Bithyniidae** | | | | |
| *Parafossarulus manchouricus* | 2,376 | 131 (5.5) | 120 | 5 (4.2) |
| *Bithynia fuchsiana* | 1,663 | 103 (6.2) | 120 | 0 |
| *Gabbia longicornis* | 318 | 0 | – | – |
| **Total** | 11,237 | 421 | 721 | 5 |

*sinensis* from other species with pleurolophocercous cercariae. *P. manchouricus* released all 7 cercariae types, while the other 4 snail species released from 1 to 5 types of cercariae.

The most common cercarial type (37.05%) of all cercariae recorded were pleurolophocercous (excluding those of *C. sinensis*), this type was found in *M. tuberculata*, *P. manchouricus* and *B. fuchsiana*. The xiphidiocercous type was the second most common (29.21%) and was released from 4 snail species. All infected snail species released echinostome cercariae which was the third most common (25.42%) type. The cercariae of *C. sinensis* composed only 2.85% of the total infections and were found only in *P. manchouricus*. Two cercariae types, furcocercous and monostome, each constituted 2.14%. Notably, the monostome type was found only in *P. manchouricus* while the furcocercous type was shed from both bithyniid species. The gymnocephalous type was the least common with a percentage of 1.19%.

Four snail species, including *M. tuberculata*, *T. granifera* (Thiaridae), and *P. manchouricus* and *B. fuchsiana* (Bithyniidae) were examined using the multiplex PCR approach for detecting

**Table 3. Cercariae types released from the first intermediate hosts.**

| Cercariae type | Snail species | | | | | No. snails (%) |
|---|---|---|---|---|---|---|
| | *M. tuberculata* | *T. granifera* | *L. swinhoei* | *P. manchouricus* | *B. fuchsiana* | |
| *Clonorchis sinensis*[*] | | | | 12 | | 12 (2.85) |
| Pleurolophocercous | 72 | | | 59 | 35 | 156 (37.05) |
| Xiphidiocercous | 48 | | 1 | 41 | 33 | 123 (29.21) |
| Echinostome | 29 | 33 | 2 | 12 | 31 | 107 (25.42) |
| Gymnocephalous | 2 | | | 3 | | 5 (1.19) |
| Furcocercous | | | | 5 | 4 | 9 (2.14) |
| Monostome | | | | 9 | | 9 (2.14) |
| Total | 151 | 33 | 3 | 131 | 103 | 421 (100) |

Note

[*] Also Pleurolophocercous.

*C. sinensis* (Table 2). Only *P. manchouricus* was found to be infected with *C. sinensis* with an overall prevalence of 4.2%. The length of the ITS1-rDNA sequence of *C. sinensis* was 271 bp while the sequence of the 28S-rDNA fragment of *P. manchouricus* was 397 bp in length. Sequences of *C. sinensis* and *P. manchouricus* were deposited to the NCBI database under accession numbers MT497281 and MT487283, respectively. Samples of three other snail species, *B. fuchsiana*, *M. tuberculata*, and *T. granifera*, were *C. sinensis* PCR negative (primers CSF and CSR) but still amplified snail DNA (primers 28SF and 28SR). The GenBank accession numbers for *B. fuchsiana*, *M. tuberculata*, and *T. granifera* are MT497280, MT497282, and MT497284, respectively.

## Experimental infection of snails by *C. sinensis*

A total of 1,450 snails of 5 species were used for experimental infection (Table 4). After 21 days post egg exposure, only *P. manchouricus* was found infected. The overall prevalence of infection was 7.87%. Additionally, cross-infected snails using eggs of *C. sinensis* from Russia and Vietnam were found. Also, of note, one multiplex PCR sample only amplified *C. sinensis* DNA and not snail DNA.

## Discussion

The presence of mollusk first intermediate hosts is the major limiting factor for the distribution of all digenean species including, *C. sinensis*. Species of Bithyniidae, which are listed as the first intermediate host of *C. sinensis*, are widely distributed [37]. Likewise, the potential second intermediate (fish) and definitive (mammals) hosts are geographically widespread, but *C. sinensis* occupies only a limited territory of South and South-East Asia. The northernmost boundary (51−52˚N latitude) of *C. sinensis* distribution coincides with that of *P. manchouricus* [38]. Due to the importance of first intermediate hosts, it is necessary to know which snail species can serve as intermediate hosts in a given area in order to implement measures to control the potential of spread of *C. sinensis*. The southernmost border of the distribution of *C. sinensis* is in the northern part of Vietnam [1,2,5] although bithyniid species, other than *P. manchouricus* are widespread throughout the whole country [39].

The successful experimental infection of *P. manchouricus* with *C. sinensis*, as well as the finding of cercariae of *C. sinensis* in this snail species in Thac Ba reservoir confirm that *P. manchouricus* is the most important first intermediate host of the Chinese liver fluke. In this study, the experimental infection rate (7.87%) was lower than that (12.5%) reported by Liang et al.

**Table 4. The snails' survival, infection after 21 days of exposure to *C. sinensis* eggs.**

| Snail species | Snail source | Eggs of *C. sinensis* from naturally infected cats (Vietnam) | | | Eggs of *C. sinensis* from experimentally infected rats (Russia) | | |
|---|---|---|---|---|---|---|---|
| | | No. snails exposed | No. snail surviving | No. snails infected (%) | No. snails exposed | No. snail surviving | No. snails infected (%) |
| **Bithyniidae** | | | | | | | |
| *B. fuchsiana* | Vietnam | 310 | 236 | 0 | 90 | 49 | 0 |
| *P. manchouricus* | Vietnam | 200 | 156 | 17 (10.9%) | 60 | 44 | 4 (9.10%) |
| | Russia | 100 | 77 | 3 (3.90%) | 40 | 28 | 0 |
| **Semisulcospiridae** | | | | | | | |
| *Koreoleptoxis amurensis* | Russia | 100 | 26 | 0 | 50 | 15 | 0 |
| **Thiaridae** | | | | | | | |
| *M. tuberculata* | Vietnam | 400 | 211 | 0 | 50 | 3 | 0 |
| *T. granifera* | Vietnam | 150 | 118 | 0 | | | |

[6]. The difference may be due to differences in the experimental conditions such as snail density, time of maintenance of snails in aquaria before experimental exposure, the maturation of miracidia in eggs, experimental duration, etc. *P. manchouricus* is a host to numerous digenean species, and experimental infections have shown it to host at least 5 trematode species in Hanoi, Vietnam, i.e. *Cyathocotyle orientalis* (furcocercous), *Echinochasmus beleocephalus* (echinostome), *Notocotylus intestinalis* (monostome), *Gymnocephala* sp. (gymnocephalous), and *Prosthogonimus* sp. (xiphidiocercous) [40]. Besprozvannykh et al. [28] also found 5 cercariae types from *P. manchouricus* in North Vietnam, namely *Echinochasmus japonicus* (echinostome), *N. intestinalis* (monostome), *Sphaeridiotrema monorchis* (gymnocephalous), *Holostephanus* sp. (furcocercous), Pleurogenidae gen. sp. (xiphidiocercous). This study only identified cercariae to major groups, but at least 7 trematode species use *P. manchouricus* as a first intermediate host. Thus, the diversity of digenean within *P. manchouricus* in this area should be studied further.

In addition, for *C. sinensis*, a cross-infection of snails from different countries (Vietnam and Russia) was obtained in this experiment. In spite of the fact that statistically significant differences were not found between parasites from Russia and Vietnam using both mitochondrial and nuclear markers [41,42], adaptations and some peculiarities are formed due to the geographic remoteness of the regions, including at the host-parasite level. Worms interacting with the hosts on a common territory are less pathogenic for them, while mollusks from another region might not have developed protective mechanisms. Thus, accidental transfer of the parasite to other regions may contribute to its rapid uncontrolled spread, since the first intermediate hosts will be more sensitive to infection.

Other bithyniid snail species such as *B. fuchsiana*, which has been confirmed as a first intermediate host of *C. sinensis* in China by the LAMP technique [8], was found uninfected with *C. sinensis* in this study. Although the LAMP technique is quite sensitive, Chen et al. [8] did not use DNA sequencing to verify their results, and only indicated that they "are able to confirm the correct products by sequencing". Moreover, as a control, they used only one representative of the family Opisthorchiidae, *Opisthorchis viverrini*, which usually does not co-occur with *C. sinensis* [1,2]. It is possible that these primers may be sensitive to other parasites occurring in the same area, for example, representatives of the genus *Metorchis*. Chung [43] also failed to infect *Bithynia misella* and *B. tentaculata* with *C. sinensis* experimentally and suggested that both species do not need to be emphasized in snail survey for clonorchiasis even though *B. misella* was reported as an intermediate host of *C. sinensis* in China [13]. The prevalence of infection of *B. fuchsiana* by trematode larvae in Thac Ba reservoir was higher than that recorded for *P. manchouricus* (6.2% vs. 5.5%). However, the prevalence of infections by cercariae of *C. sinensis* was contrasting, 9.16% (12/131) in *P. manchouricus* and 0% in *B. fuchsiana*. Multiplex PCR also did not detect *C. sinensis* from 120 specimens of *B. fuchsiana*, which suggests that the prevalence of infection of *B. fuchsiana* in the wild is very low or *B. fuchsiana* is not a suitable host of *C. sinensis* in this region.

In Vietnam, the first report of *M. tuberculata* as the first intermediate host of *C. sinensis* was published by Kino et al. [20], and the prevalence of infection was 13.3% in Ninh Binh province. De [21] reported a prevalence of *C. sinensis* infections in *M. tuberculata* of 10.2% in the Red River Delta. Bui et al. [26] did not find any cercariae of *C. sinensis* from 3,335 specimens of *M. tuberculata* and 34 specimens of *T. granifera* in Nam Dinh province while prevalence of infection with small intestinal trematode larvae was high. Similarly, Nguyen et al. [27] also did not find any *C. sinensis* cercariae from 758 specimens of *M. tuberculata* in Ninh Binh province which is the most important endemic area for clonorchiasis in Vietnam [21]. The contrasting results may be based on misidentification of cercariae. Cercariae of *C. sinensis* belong to "pleurolophocercous" type. This type is found within three families, i. e. Opisthorchiidae,

Cryptogonimidae, and Heterophyidae [44,45]. Bui et al. [26] found that parapleurolophocercous (cercariae of the subfamily Haplorchiinae) constituted 40.3% of all infections found in snails while pleurolophocerca only constituted 0.3% in Nam Dinh province, an endemic area of *C. sinensis*. Pinto [45] suggested that of the term "parapleurolophocercous" should no longer be used and these cercariae should be referred to as pleurolophocercous. Moreover, if *M. tuberculata* was the first intermediate host of *C. sinensis*, it would be difficult to explain why *C. sinensis* only occurs in northern Vietnam while its intermediate host, *M. tuberculata*, is distributed throughout the entire country. The failure to infect *M. tuberculata* and *T. granifera* by *C. sinensis* in this study suggests that these two snail species should be removed from the list of first intermediate hosts of *C. sinensis*.

*Koreoleptoxis amurensis* is distributed throughout the endemic areas of *C. sinensis* in Russia [7,46]. This snail species has not been reported to be infected with *C. sinensis*, however it has been found shedding cercariae of many trematode species, e.g. *Nanophyetus salmincola* (Nanophyetidae), *Plagioporus* sp. (Opecoelidae), *Sanguinicola* sp. (Sanguinicolidae), *Echinochasmus* sp., *Microparyphyim* sp. (Echinostomatidae), *Metagonimus* spp., *Pygidiopsis* sp., and *Centrocestus* (Heterophyidae), and xiphidiocercous cercariae [44]. In addition, this snail belongs to the family Semisulcospiridae, representatives of which were also listed as first intermediate hosts of *C. sinensis* in China [14,22]. The failure to infect *K. amurensis* by *C. sinensis* in this study confirms that this snail species is not a suitable host for *C. sinensis*.

The multiplex PCR approach has been used for the detection or the differential diagnosis of *C. sinensis* and other trematodes [47,48]. This method, however, has not been reported to detect *C. sinensis* DNA from within its host snail. The present study provides a novel multiplex PCR for amplification of multiple genes of both snail and *C. sinensis* DNA. This is the first report of using a multiplex PCR approach to detect larvae of *C. sinensis* within their snail first intermediate host. Prevalence of infection by *C. sinensis* within *P. manchouricus* in Thac Ba reservoir, detected by multiplex PCR, was 4.2%, which is nearly 1.5 times higher than the prevalence detected by observation for cercariae shedding (2.85%).

The results obtained can be used to clarify the list of the first intermediate hosts for *C. sinensis* throughout its range and show that in the case of studying any aspect related to trematodes, it is necessary to correctly identify not only the parasite species but also their hosts. Since wrong data incorrectly indicate the epidemiological potential of any parasite, which can lead to sudden unpredictable outbreaks of diseases in the regions. In addition, other situations possible when the spread of trematodes may be expected in the region due to erroneous identification of the parasite. For example, parasites that were previously recorded in the mollusks *M. tuberculata* and *T. granifera*, not being *C. sinensis*, in the presence of all conditions for life cycle, can continue to spread in the region without control until they reach a critical threshold. In both cases, it affects the wrong investment in the development of preventive, diagnostic and therapeutic measures that contribute to the health of the population.

In conclusion, we have shown, through both examination of naturally infected snails, as well as experimental infections, that in Northern Vietnam the two thiarids, *M. tuberculata* and *T. granifera* are not suitable first intermediate hosts of *C. sinensis*. Other snail species, which were reported as the first intermediate host of *C. sinensis*, should be reconfirmed by molecular data and/or experimental infection.

## Acknowledgments

The authors are most grateful to Dr. Stephen E. Greiman, Georgia Southern University, for his careful proof reading of this manuscript.

## Author Contributions

**Conceptualization:** Hung Manh Nguyen, Henry Madsen.

**Data curation:** Hung Manh Nguyen.

**Formal analysis:** Hung Manh Nguyen, Yulia V. Tatonova.

**Funding acquisition:** Hung Manh Nguyen.

**Investigation:** Hung Manh Nguyen, Hien Hoang Van, Loan Thi Ho, Yulia V. Tatonova, Henry Madsen.

**Methodology:** Hung Manh Nguyen, Loan Thi Ho, Yulia V. Tatonova, Henry Madsen.

**Project administration:** Hung Manh Nguyen.

**Supervision:** Henry Madsen.

**Validation:** Hung Manh Nguyen, Yulia V. Tatonova.

**Writing – original draft:** Hung Manh Nguyen.

**Writing – review & editing:** Hung Manh Nguyen, Yulia V. Tatonova, Henry Madsen.

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
