## [Decision Letter · Decision Letter 0]

17 Sep 2020

Dear Mr. Nguyen,

Thank you very much for submitting your manuscript "Are the thiarids, Melanoides tuberculata and Tarebia granifera, first intermediate hosts for Clonorchis sinensis?" for consideration at PLOS Neglected Tropical Diseases. As with all papers reviewed by the journal, your manuscript was reviewed by members of the editorial board and by several independent reviewers. In light of the reviews (below this email), we would like to invite the resubmission of a significantly-revised version that takes into account the reviewers' comments. 

We cannot make any decision about publication until we have seen the revised manuscript and your response to the reviewers' comments. Your revised manuscript is also likely to be sent to reviewers for further evaluation.

Sincerely,

Xiao-Nong Zhou

Associate Editor

Banchob Sripa

Deputy Editor

Reviewer's Responses to Questions

**Key Review Criteria Required for Acceptance?**

**Methods**

-Are the objectives of the study clearly articulated with a clear testable hypothesis stated?

-Is the study design appropriate to address the stated objectives?

-Is the population clearly described and appropriate for the hypothesis being tested?

-Is the sample size sufficient to ensure adequate power to address the hypothesis being tested?

-Were correct statistical analysis used to support conclusions?

-Are there concerns about ethical or regulatory requirements being met?

Reviewer #1: the method section is good. All the criteria were met. I just suggest to add more experiments to the paper.

Reviewer #2: The objectives and study design are appropriate for the study objectives, however, more details on certain aspects of the study design are needed to assess if the experimental infections were conducted in a manner that gives accurate results. Please see the method section comments in the attached Reviewer's Comments document.

Reviewer #3: The Methods are clear, I only have one question:

why were the snails fixed and not dissected alive for the trematodes? it is easier to dissect live snails

**Results**

-Does the analysis presented match the analysis plan?

-Are the results clearly and completely presented?

-Are the figures (Tables, Images) of sufficient quality for clarity?

Reviewer #1: The tables are fine but the figure needs better resolution

Reviewer #2: The analysis match the descriptions in the methods. The manuscript has a lot of tables and the authors may want to consider making one or two of these tables into graphs to add the reads in understanding the data.

Reviewer #3: Concerning the Figure, it would be helpful to accompany it with a map of whole Vietnam and to indicate where the region under study is.

**Conclusions**

-Are the conclusions supported by the data presented?

-Are the limitations of analysis clearly described?

-Do the authors discuss how these data can be helpful to advance our understanding of the topic under study?

-Is public health relevance addressed?

Reviewer #1: Yes

Reviewer #2: Overall the discussion needs major revisions to better explain the findings and their importance as well as make the comparisons to other research more informative (instead of reading like a list of previous paper's findings). The conclusions are supported by the data, but are extended to a wider geographic area than the data support. Thus, the authors need to make revisions to the language to suggest that their finding may apply to a broader area instead of being definitive that their results are true for all locations. The discussion would be improved with a few paragraphs that explain the implications both ecologically and on human health.

Reviewer #3: The conclusions are supported by the data, however, the authors should more emphasise the need of correct identification of parasites and hosts.

**Editorial and Data Presentation Modifications?**

Reviewer #1: Please check the uploaded revised version of the manuscript.

Reviewer #2: (No Response)

Reviewer #3: The text needs a native English speaker for language checking, also, please check your text for spelling errors:

In the affiliation (line 10) please spell Vladivostok correctly

Running title – please reformulate the title, it is not understandable

In the Introduction (line 56) for Clonorchis sinensis in the first sentence, please add the nomenclatural author

line 59

please correct ‘multipleabscesses’ to ‘multiple abscesses’

Table 1

it would be very helpful to provide the snail species names with nomenclatural authors and spell out in full the generic names

please correct ‘Semisulcospira libertine’ to ‘Semisulcospira libertina’

line 112

please correct to ‘Experimental infection of snails’

line 123

please correct ‘sisnenis’ to ‘sinensis’

line 129

please correct ‘tape’ to ‘tap’

line 174

please correct ‘from 4 snail’ to ‘in 4 snail’

lines 175-176

please reformulate the sentence ‘The cercariae of C. sinensis possessed only 2.85% of total

176 infected snails.‘

line 176

please correct ‘This cercariae’ to ‘These cercariae’ 

line 193

please correct ‘infection for snails’ to ‘infection of snails’

line 198

please correct ‘survive snails’ to ‘survived snails’ 

there are several more things to be corrected in the Discussion

**Summary and General Comments**

Reviewer #1: PNTD-D-20-00929 

Title

Are the thiarids, Melanoides tuberculata and Tarebia granifera, first intermediate hosts for Clonorchis sinensis? 

This manuscript by Nguyen et al. study the potential of two thiarid snails; Melanoides tuberculata and Tarebia granifera as intermediate hosts for the Chinese liver fluke Clonorchis sinensis. They collect the snails from a parasite endemic area in Vietnam and checked it for natural infection using traditional and molecular techniques. In addition, they tried to infect the same snails from parasite free areas with C. sinensis. Based on their field data and laboratory infection experiments they concluded that these two thiarids are not potential hosts for C. sinensis and Parafossarulus manchouricus is the only molluscan intermediate host for this trematode. 

General comments:

The authors did not consider that these two thiarid snails might be incompatible due to some environmental factors because they were collected from the field. Generally, it is difficult to infect field captured snails? May be they can try with their lab-bred snails.

More experimental work can be done to detect the fate of the parasite within resistant snails such as by using histopathological sectioning of the head-foot.

A better resolution of the map is needed.

The language of the manuscript needs some improvement.

Please revise the scientific names of the snails and trematode species.

More comments and edits are included in the uploaded version of the manuscript.

Reviewer #2: The comments in this section can also be accessed in the Reviewer's Comments document.

Dear Authors,

Overall, I found your research interesting and the multiple approaches to assess the hosts for C. sinensis quite sufficient. However, I do have a few concerns regarding the manuscript which are explained below.

Overall

• All parts of the manuscript would be improved by changing at least half the sentences in the paragraphs to the active voice. I will provides some examples of sentences I would change to the active voice in the comments for each section.

• Some of the paragraphs throughout the manuscript are not indented. If this is required for the journal than nothing needs to be done, but please check to ensure that you are consistent in the use of indenting for the start of paragraphs.

• In multiple places in the manuscript, the authors assert that their findings for a single reservoir are applicable to a larger geographic area. Although their finding may suggest that the two snails are not hosts in a larger area, the wording needs to be less certain to accurately portray the research. I will provide specific comments in the section this needs to be applied, but the authors should check the entire manuscript to ensure that they are not overstating their findings.

Title

• The word “Precise” in the Running Title should be replaced with a verb. For example, “Assessing” or “Evaluating” would make the Running Title easy to understand.

Abstract

• Lines 22-23:Please rewrite the second sentence to increase clarity. One possibility would be to change the sentence to read: “However, theses snails’ role as intermediate hosts has not been confirmed.” This simplifies the sentence and increases clarity.

• Lines 23-27: In the third sentence that begins: “Here we present” change “Here” to “Thus,” as a better transition. Also in this sentence change “firstly” to “first” and end the sentence after the word reservoir The new sentence that begins on line 27 would then read “ Next, we attempted to infect these snails with eggs of C. sinensis in laboratory experiments.”

• Lines 32-34: The information on the accession numbers is out of place in the abstract. Unless required to be in the abstract by the journal, move this information to the methods and delete from the abstract.

• Line 38: add “in the Thac Ba reservoir” before “in Vietnam” to make this statement more accurate as your findings only apply to this area and not the entire country.

Authors’ Summary

• Lines 46-47: rewrite the sentence for clarity to read “Both snail species have wide distributions in tropical and subtropical waters across the world, but C. sinensis only occurs in East Asia.”

• Line 48: in the second half of the sentence change “we confirmed that in Vietnam” to “confirms that in Northern Vietnam”

• Line 50: change “detected in Parafossarulus manchouricus by observation, molecular analyses and experimental infection” to “detected in a single snail species, Parafossarulus manchouricus, through observations, molecular analyses, and experimental infections.”

Introduction

• Line 58-60:first change “e.g.” to “including”; then remove “it from the sentence; then add “a” before “high number of human” and finally add “of infection” to the end of the sentence.

• Line 70: change the sentence to read “Melanoides tuberculate and T. granifera are highly invasive species and are currently present . . .”

• Lines 83-84: remove “e.g.” and use colon after the word “approaches” then add a comma after “experimental infection”

Materials and Methods

• Line 92: add an “s” to latrine

• Lines 93-94: place “e.g. cattle, pig, ducks, chickens, dogs, cats in parentheses

• Line 98: change “collecting” to “collection”

• Line 99: change “each four” to “every four”

• Line 112: change “for” to “of snails”

• Lines 115-116: I rewrote this sentence to be in the active voice: “We collected snails from trematode-free locations and acclimatized them to laboratory conditions for seven days.” In the rest of the methods section change roughly half of the sentences to the active voice to increase readability of the manuscript.

• Lines 118-122: The information in this paragraph is difficult to follow. I attempted to rewrite it as follows: “For maintenance of experimental snails, we filled aquaria (plastic boxes with 2 L capacity) with dechlorinated tap water and a bottom layer of sand and small stones. To mimic the natural habitat of the snails, we add a few pieces of brushwood (Terminalia cattappa) and aquatic plants (Hygrophila difformis, Lemna minor) to each aquarium. We housed ten individuals of the same snail species in each aquarium. The number of each snail species used in this experiment varied (Table 4).

• Line 122: The reference to Table 4 places the tables out of order. Please check the journal guidelines to determine if you need to reorder the tables to make them sequential in the manuscript.

• Lines 123-131: The collection methods for C. sinsnesis adults and eggs need to be expanded to answer the following questions: How were cats sampled? How were eggs quantified or were large amounts just added to each aquarium? How did you ensure that eggs were present in all 5 mL allocates put into aquaria? These questions need to be addressed to better understand the experimental design and to ensure that all aquaria were actually infected with C. sinsnesis eggs.

• Lines 123-126: The sentences on adult flatworm collection are difficult to read – Rewrite to include answers to the previous comments’ questions.

• Lines 128-131: rewrite the sentence to “20 grams of feces from domestic cats (free from parasites) to 700 ml tap water which was shaken until the fecal matter disintegrated.”

• Lines 132-132: This sentence needs to be incorporated into the previous paragraph and not its own paragraph.

• Lines 151 and 156: Should these paragraphs be indented? Also, do these need to be new paragraphs?

Results

• Lines 171-172: This sentence needs to be rewritten to explicitly state the C. sinensis is a pleurolophocercous cercariae. As written, it took a lot of effort including looking at the tables and search for C. sinensis on Google to determine that C. sinensis is a part of pleurolophocercous cercariae and that the authors are grouping them separately. 

• Lines 173-174: remove the comma after (29.21%)

• Lines 174-175: change “cercariae, and this type” to “which was the third most common (25.42) cercarial type.”

• Lines 175-176: change “infected snails. This cercariae” to “infected snails and were”

• Line 179: add “cercarial type” after gymnocephalous

• Line 185: table should be capitalized.

• Lines 196-197: The last two sentences in the results are difficult to read and do not make sense to me. Please rewrite these sentences for clarity.

Discussion

• Overall, the discussion reads like a list of others’ findings with minimal comparisons to the finding in this manuscript. The discussion needs to be re-written to focus on the findings of this manuscript and their importance and comparison to past studies. As currently written, it is difficult to follow the reasoning or main ideas of the discussion and the conclusions are overly confident and applied to quickly to larger regions considering the research in the manuscript comes from a single reservoir. I would except more discussion on alternative explanation for why others found evidence for these two snails being the host of C. sinsnesis.

• Given that substantial revisions of the discussion are needed, I will only point out a few other issues in this sections:

o Line 202: remove “the” from in front of Bithyniidae

o Line 205: add “distribution” after C. sinensis

o Line 229: change “being” to “as”

o Lines 231-233: this sentence is extremely difficult to understand.

o Line 273: C. sinensis needs to be italicized

o Lines 282-284: rewrite the sentence in the active voice: “In this study, we report the first use of 28S-rRNA gene of M. tuberculate, T. granifera, B. fuchsiana and P. machouricus form Vietnam.”

o Lines 285-287: This sentence is too strong of a conclusion. You did not sample locations throughout Vietnam and thus need to rewrite to be less assertive on your claims.

Tables and Figures

• Table 1 – In the first column, move the asterisks and ~ to after the names of the snail species to make these less distracting to the reader (unless this position in the table is required by the journal).

• Tables 3 and 4 --- You may want to consider making one or both of these tables into graphs to better show the data.

• Table 3 – in the footnote for this table there is a ) that seems out of place

Reviewer #3: The authors examine the ability of two snail species to host Clonorchis sinensis; it is a serious problem, as the snails are widely distributed, therefore, the paper is of importance and it is exactly within the scope of the Journal. 

Most of the approach of the authors is logic, however, it is not clear already in the Abstract why they sampled also other snails and why they provide sequences of the snail hosts. This is an important addition to the topic, but it should be explained, as they did not examine only Tarebia granifera and Melanoides tuberculata. Also the title does not fully correspond to the contents, as it makes the reader expect that only the two snail species will be treated, however, the scope of the MS is wider.

It would be wise to emphasise in the text that it is important to correctly identify the snail species and also the parasite species, as C. sinensis has pleurolophocercariae which are quite uniform in most species and they are typical in other medically important trematode families. The authors treat this in the Discussion, but it would be good to emphasise the importance of correct identification already in the Abstract, also the use of Multiplex PCR.

PLOS authors have the option to publish the peer review history of their article (what does this mean?). If published, this will include your full peer review and any attached files.

Reviewer #1: No

Reviewer #2: No

Reviewer #3: No
---

## [Decision Letter · Decision Letter 1]

15 Nov 2020

Dear Mr. Nguyen,

Thank you very much for submitting your manuscript "Are Melanoides tuberculata and Tarebia granifera (Gastropoda, Thiaridae), suitable first intermediate hosts of Clonorchis sinensis in Vietnam?" for consideration at PLOS Neglected Tropical Diseases. As with all papers reviewed by the journal, your manuscript was reviewed by members of the editorial board and by several independent reviewers. The reviewers appreciated the attention to an important topic. Based on the reviews, we are likely to accept this manuscript for publication, providing that you modify the manuscript according to the review recommendations. 

Sincerely,

Xiao-Nong Zhou

Associate Editor

Banchob Sripa

Deputy Editor

Reviewer's Responses to Questions

**Key Review Criteria Required for Acceptance?**

**Methods**

-Are the objectives of the study clearly articulated with a clear testable hypothesis stated?

-Is the study design appropriate to address the stated objectives?

-Is the population clearly described and appropriate for the hypothesis being tested?

-Is the sample size sufficient to ensure adequate power to address the hypothesis being tested?

-Were correct statistical analysis used to support conclusions?

-Are there concerns about ethical or regulatory requirements being met?

Reviewer #1: The objectives are clear and the study design is appropriate.

Reviewer #2: Overall the study objectives and design are adequate, but a few questions still need to be addressed:

Why were snails from Russia included in this study? You need to explain the reasoning behind adding the Russian snails since the rest of the paper focuses on Northern Vietnam.

The collection methods for C. sinsnesis adults and eggs need to be expanded to answer the following questions: How were cats sampled? What permits if any were used for dealing with the cats? How were eggs quantified or were large amounts just added to each aquarium? How did you ensure that eggs were present in all 5 mL allocates put into aquaria? These questions need to be addressed to better understand the experimental design and to ensure that all aquaria were actually infected with C. sinsnesis eggs.

**Results**

-Does the analysis presented match the analysis plan?

-Are the results clearly and completely presented?

-Are the figures (Tables, Images) of sufficient quality for clarity?

Reviewer #1: Yes. All the criteria were met.

Reviewer #2: The results are well described in the text, but the figure legends and table captions need more detailed explanations.

**Conclusions**

-Are the conclusions supported by the data presented?

-Are the limitations of analysis clearly described?

-Do the authors discuss how these data can be helpful to advance our understanding of the topic under study?

-Is public health relevance addressed?

Reviewer #1: Yes the conclusion supported the data.

Reviewer #2: Overall, the discussion is much improved and easier to read, but the first paragraph of the discussion would work better as the second paragraph of the introduction. 

You may want to add a sentence or two at the end of the discussion to explain the significance of the study to human health and preventing human disease due to the trematode.

**Editorial and Data Presentation Modifications?**

Reviewer #1: PNTD-D-20-00929R1

Title 

Line 2-3: for Clonorchis sinensis in Vietnam? – for instead of “of” - consider this in the entire manuscript - and in Vietnam is not italic

Abstract

Line 27-28: As well as, through experimental infections of five these snail species with eggs of C. sinensis. Remove five!

Line 30: Five snail species; …….. mention these species

Line 34: infective C. sinensis eggs…… remove infective

Line 35: infected, at an infection rate of 7.87%. with an infection rate.

Line 38: hosts

Keywords: small liver fluke

Author summary

Line 45: The role that these two thiarid snail species play.

Introduction

Table 1: sometimes you put the name of the one who first described the species between brackets and sometimes not. Please unify. 

Line 73: Mas-Coma & Bargues [1] mentioned that their wide distribution potentially could result in the global.

Materials and Methods

Line 113: Five snail species were chosen for experimental infection. Based on what?

Line 130: Eggs were placed in 20 g of feces.

Results

Line 165: A total of 11,985 snails, representing 10 species, were collected (Table 2). You already mentioned that in the M&M.

Line 166: snails were examined for shed cercariae cercarial shedding

Line 187: Table 3. Cercariae 

The last column on the right No snails (%). Do you mean No of Cercariae (%)?

Line 205: Table 4. Can you make the table legend more clear? The language is not clear. 

You can transfer your data to describe the snails’ survival, infection and mortality rates after 21 days of exposure to C. sinensis eggs!

Discussion

Line 227: P. manchouricus is a host….

Line 286: muntiplex PCR

Reviewer #2: The captions for the tables and graphs only have single sentence titles and lack detailed explanations. These table captions and figure legend need to be expanded to be more accurate and provide more information about what is shown in the graph and tables.

Table 3 or 4 could be changed to figures to make the information in these tables easier to access for the reader.

**Summary and General Comments**

Reviewer #1: Thanks for addressing the previous comments on the first submission. I encourage the authors to further read and revise their manuscript to reach the optimum before publication. 

Nice work and congratulations!

Reviewer #2: Overall, I found your manuscript much improved and easier to read, however, I still have a few concerns (see Reviewer Comments document).

PLOS authors have the option to publish the peer review history of their article (what does this mean?). If published, this will include your full peer review and any attached files.

Reviewer #1: No

Reviewer #2: No
---

## [Decision Letter · Decision Letter 2]

3 Jan 2021

Dear Mr. Nguyen,

We are pleased to inform you that your manuscript 'Are Melanoides tuberculata and Tarebia granifera (Gastropoda, Thiaridae), suitable first intermediate hosts of Clonorchis sinensis in Vietnam?' has been provisionally accepted for publication in PLOS Neglected Tropical Diseases.

Best regards,

Xiao-Nong Zhou

Associate Editor

Banchob Sripa

Deputy Editor

Reviewer's Responses to Questions

**Key Review Criteria Required for Acceptance?**

**Methods**

-Are the objectives of the study clearly articulated with a clear testable hypothesis stated?

-Is the study design appropriate to address the stated objectives?

-Is the population clearly described and appropriate for the hypothesis being tested?

-Is the sample size sufficient to ensure adequate power to address the hypothesis being tested?

-Were correct statistical analysis used to support conclusions?

-Are there concerns about ethical or regulatory requirements being met?

Reviewer #1: The methods section is appropriate and regulatory requirements were met.

Reviewer #2: The changes to the methods now incorporate the information on how adult parasites were collected and the permits required for IACUC or similar ethical care of animals.

**Results**

-Does the analysis presented match the analysis plan?

-Are the results clearly and completely presented?

-Are the figures (Tables, Images) of sufficient quality for clarity?

Reviewer #1: Yes

Reviewer #2: The results and tables are adequate.

**Conclusions**

-Are the conclusions supported by the data presented?

-Are the limitations of analysis clearly described?

-Do the authors discuss how these data can be helpful to advance our understanding of the topic under study?

-Is public health relevance addressed?

Reviewer #1: Yes

Reviewer #2: The rewritten discussion and conclusion are supported by the data and show improved clarity for the importance of this research.

**Editorial and Data Presentation Modifications?**

Reviewer #1: There are a few minor edits in the attached doc version of the manuscript

Reviewer #2: Accept

**Summary and General Comments**

Reviewer #1: The authors addressed the issues raised by the reviewers and the publication merits manuscript in PLoS NTD.

Reviewer #2: Overall the manuscript is much improved from the original.

PLOS authors have the option to publish the peer review history of their article (what does this mean?). If published, this will include your full peer review and any attached files.

Reviewer #1: No

Reviewer #2: No

---

## [Editor Report · Acceptance letter]

21 Jan 2021

Dear Mr. Nguyen,

We are delighted to inform you that your manuscript, "Are Melanoides tuberculata and Tarebia granifera (Gastropoda, Thiaridae), suitable first intermediate hosts of Clonorchis sinensis in Vietnam?," has been formally accepted for publication in PLOS Neglected Tropical Diseases.

Best regards,

Shaden Kamhawi

co-Editor-in-Chief

Paul Brindley

co-Editor-in-Chief
